# Psychometric Properties of the Self-Healing Assessment Scale for Community-Dwelling Older Adults

**DOI:** 10.3390/healthcare9040484

**Published:** 2021-04-20

**Authors:** Yi-Chen Wu, Hua-I Hsu, Heng-Hsin Tung, Shi-Jun Pan, Shu-Wei Lin

**Affiliations:** 1College of Nursing, National Yang Ming Chiao Tung University, Taipei 112, Taiwan; yichenwu.y@nycu.edu.tw; 2Department of Nursing, Fu Jen Catholic University Hospital, Fu Jen Catholic University, Taipei 243, Taiwan; 3Department of Educational Psychology and Counseling, School of Education, National Taiwan Normal University, Taipei 106, Taiwan; 4Yang Sheng Foundation, Taipei 106, Taiwan; pandora@ysfoundation.org.tw (S.-J.P.); bookwai@ysfoundation.org.tw (S.-W.L.)

**Keywords:** self-healing, older adults, psychometric properties, nursing, instrument development

## Abstract

Self-healing, an intrinsic healing capacity, helps individuals’ bodies and minds to regain wholeness and is significant in the pursuit of one’s own healthy ageing and independence. This study was intended to develop and preliminarily test the reliability and validity of the self-healing assessment scale (SHAS) for community-dwelling older adults, and was conducted in three phases. Phase 1: The definitions of self-healing were synthesized from our knowledge of the literature regarding the ontology of self-healing and panels of 25 experts. The initial version of the 12-item questionnaire was developed by the in-depth interviews of focus groups and panels, and the content was validated by six experts. Phase 2: A cross-sectional survey, including a total of 500 community-dwelling older adults with a mean age of 71.76, was then conducted for the preliminary reliability and validity test. The content validity indices were satisfied. Twelve items were retained, and three factors were identified, namely, physical and mental state, socioeconomic and environmental status, and independent lifestyle, which explained 65.8% of the variance under explorative approval. Phase 3: the standardized factor above 60 obtained by confirmatory factorial analysis indicated good convergent validity. The relationship between self-healing and health-related quality of life was confirmed via concurrent validity testing. The SHAS can facilitate the evaluation of factors associated with community-dwelling older adults’ self-healing capacity. Programs tailored to enhance self-healing capacity should be designed, implemented, and inspected regarding their effectiveness in older adults.

## 1. Introduction

With life expectancy increasing and the older population growing up to 22% in 120 million by 2050 worldwide, aging is now attracting focused attention through research for producing long-term positive outcomes in gerontology [1,2]. The inclusive concept of healthy aging is defined by the World Health Organization (WHO) as the process of developing and maintaining the functional ability that enables wellbeing in older age, and it captures the essence of physical and cognitive functional preservation, including functional ability and intrinsic capacity maximization, but excludes the requirement of disease avoidance [3,4,5].

As people age, general health status deteriorates, leading to a gradual decrease in physical and mental capacity, a decline in functioning and independence, a growing risk of contracting diseases, and ultimately death [6,7]. Social experiences also change with age. For instance, social isolation and loneliness; experiencing the death of friends and/or partners; and life transitions, such as retirement and relocation, which cause negative health outcomes and early mortality, have been well documented in studies [2,6,8]. Thus, age-related decline, causing inactivity, loss of independence, and a shrink in social network affects individuals, families, and societies [6,7,9]. To achieve healthy aging, it is important not only to reduce losses associated with older age but also reinforce recovery, adaptation, and psychosocial growth through a process in which biological, lifestyle, and environmental factors interact over time to produce long-term positive outcomes [1,2,10].

In health care, healing has been defined as the ability to endure and reach homeostasis, a unification of mind, body, and spiritual wellbeing, and an improvement in quality of life (QOL) for people undergoing unfavorable circumstances [11,12,13]. A shift toward healing and QOL has been illustrated recently and depends on the choice for a new approach to practice and an embrace of holistic care using tools common in the prevailing paradigm [11,12]. Additionally, healing is seen as traces of a transcendent or encompassing wholeness that involves physical, mental, emotional, social, and spiritual aspects of human experience in which wholeness is grounded in human life [14,15]. Self-healing, an intrinsic healing capacity, acts as the process of rebalancing to regain wholeness and is significant in pursing one’s own health by giving rise to an outcome of transcendence under a holistic approach [16,17,18,19]. Processes of self-healing have been promoted by physicians for eliminating the symptoms of illness, in which the capacity of self-healing is the fundamental change between the two opposites of health and illness [13,20]. To govern the process of self-healing, it is necessary to study it to develop interventions for healthy aging.

The definitions of self-healing were synthesized from our knowledge of the literature regarding the ontology of self-healing and panels. A systematic review of published studies was performed, and three bibliographic databases (PubMed/Medline, ProQuest, and Ovid) along with Google Scholar were used to search and retrieve articles published up to May 2018. The following Boolean expression was used: “(heal OR healing) AND (self-heal OR self-healing)”. Fábrega maintained that “sickness and healing was a natural tendency for problems of adaptation to be manifested in terms of physiology, emotional experience, and behavior” [21]. Nightingale’s healing from a holistic perspective is defined as a process of bringing together all aspects of one’s body, mind, and spirituality to achieve and maintain integration and balance [22]. McKie identified healing, through a whole-person care approach in nursing practice, as a natural and active capacity that occurs from within to restore body homeostasis, to self-diagnose and repair, and is presented as a responsibility for one’s own health [23]. Egnew claimed that healing was the personal experience of transcending suffering [15], whereas Mount et al. expressed healing as a unification of mind, body, and spiritual wellness [24]. In Watson’s caring–healing theory for older adults, healing was stated as the interconnectedness of the body, mind, and spirituality. Moreover, the environment could influence one’s sense of connectivity, energy, and inner harmony [25]. McElligott applied healing in nursing practice and defined it as a positive, subjective, unpredictable process, involving a transformation to a new sense of wholeness, spiritual transcendence, and reinterpretation of life [26]. Firth et al. defined healing as a “holistic, transformative process of repair and recovery in mind, body, and spirit, resulting in a positive change, finding meaning, and movement toward self-realization of wholeness, regardless of the presence or absence of disease [27]”. Accordingly, self-healing was maintained by Robb within the context of complementary therapy, and described as “an active, innate, and personal process that, upon the use of an energetic catalyst, results in the rechanneling of innate and vital energy forces throughout the journey toward transcendence” [18].

In preventive medicine, behavioral health interventions of self-healing that integrate mindfulness practice based on the Indo-Tibetan tradition have been shown to reduce stress and enhance learning and QOL in breast and gynecologic cancer survivors [28,29]. The contemplative self-healing methods of meditation-based stress reduction programs, healing visualization-based health education, guided through exercises emphasizing breathing awareness, healing imagery, and deep breathing to gain better control in responses and to practice self-change, showed improved QOL and greatly reduced posttraumatic stress disorder symptoms in breast cancer survivors [30,31]. Various methods, for instance, hypnosis, with training in adaptation and relaxation as biobehavioral strategies and nature-based environments, including outdoor gardens, green spaces, parks, or forests, have been used to modulate the capacity of healing in restoring either physical or mental health with immunity generally [18,19,23,32,33]. Interestingly, laughter therapy, serving as a healing agent, has been shown to reduce chronic pain and feelings of loneliness and increase happiness and life satisfaction in older adults and their families [34].

The capacity of self-healing in older adults is especially crucial for functional independence; findings show that enhancing independence in self-care activities improves older adults’ QOL and alleviates the caregiver [20,35]. Self-healing to help older adults access the ability to perform activities of daily living (ADL), including grooming and/or personal hygiene, dressing, toileting and/or continence, transferring and/or ambulating, eating, and engaging social activity, could give rise to positive outcomes in healthy aging [35]. Additionally, exercise, healthy diet and lifestyle, and interpersonal interactions have been shown to reduce or delay physical and psychological illness in older adults [34,36], but the mechanisms are unknown. The efficacy of self-healing intervention programs in previous studies, however, had a psychosocial impact on the disease, using the Impact of Events Scale (IES), QOL outcomes, and the general Functional Assessment of Cancer Therapy Scale (FACT-G) [30,31]. Therefore, the purpose of this study was to develop and examine some initial aspects of the reliability, factor structure, and validity of the self-healing assessment scale (SHAS) in assessing the self-healing capacity of community-dwelling older adults.

## 2. Methodology

The study was designed in accordance with the recommendations for scale design and development, which include three phases: item development (phase 1), scale development (phase 2), and scale evaluation (phase 3) [37].

### 2.1. Instruments and Procedure

#### 2.1.1. Phase 1: Item Development of the Initial Version of SHAS

Based on the literature review, the self-healing definition in Robb, particularly, is not acceptable and feasible in healthy aging. Self-healing was expected to be a difficult concept to define and discuss, particularly for healthy aging. Panels, containing 25 experts in medicine, geriatrics, nursing, physical therapy, occupational therapy, nutrition, physical education, sociology, psychology, philosophy of cognitive neuroscience, immunology, linguistics, and biostatistics profession, began with several focus group discussions based on the recognition of a broad variety of professionals interested in self-healing; they defined self-healing in older adults as a natural-born healing capacity that coordinates physiological system functions smoothly and keeps our bodies and minds in a balanced state, which were the basic criteria for item generation and were conceptualized as the need for a self-report questionnaire with contents spanning these domains.

Three major domains contained 12 questionnaire items, which were physical status (4 items), mental state (3 items), social and environment (4 items), and 1 specific item about self-healing characterized by in-depth interviews of focus groups and panels, respectively. The initial items of SHAS generated by two panels and a preliminary questionnaire of SHAS were generated and modified through item revision and selection in the further three panels for applying a face and content analysis and constant comparison methods. Face validity was assessed through open-ended questions that were posed to the experts in exploring the understanding of the items and their views on the overall concept about the design of scale. Content validity, conversely, was assessed through a quantitative agreement of the experts regarding their recognition of the relevance of each item by using two Likert scales that ranged from 1 (completely not relevant) to 2 (not relevant), as one group, and 3 (relevant) to 4 (completely relevant), as another group, for the content validity index (CVI) calculation. With six experts, an item-level CVI (I-CVI) of 0.78 or higher indicated excellent content validity [38]. The quality of items was improved through item revision and two pilot studies, which occurred as the preliminary examination of response to items on SHAS. The respondents rated each item on a 4-point scale, with 4 = totally agree, 3 = agree, 2 = disagree, and 1 = totally disagree. Higher scores indicated higher self-healing capacity. A 4-point Likert scale was the response option selected to avoid a neutral midpoint and increase the validity of the finding in SHAS in an older population [39,40]. The Cronbach’s Alpha in two pilot studies was 0.70 and 0.90 sequentially under 30 and 87 participants, respectively. The SHAS did not have any items that needed to be reduced of this process. Finally, an initial 12-item version of SHAS was used to explore the factor structure in a much larger sample size and conduct explorative approval and confirmatory factorial analysis (CFA) [41,42].

#### 2.1.2. Phase 2: Scale Development for Examining the Reliability and Validity of SHAS

After the initial version of the SHAS was generated, cross-sectional data collection using a convenience sampling approach was performed in the community. Explorative approval included item analysis and exploratory factorial analysis (EFA). Descriptive statistics as mean and standard deviations (SD) were used to synthetize the demographic characteristics and the items’ distribution after data collection. For continuous normally distributed variables, which were presented by the mean and SD, skewness/standard error (SE) skewness, kurtosis/SE kurtosis, and comparisons of extreme groups were evaluated to ascertain the normality and discrimination. For the homogeneity test, interitem correlation was analyzed by Pearson’s correlation in which a coefficient between 0.3 and 0.7 indicated good item redundancy [43]. The factorability of the correlation matrix and sample adequacy-to-factor analysis were inspected using Bartlett’s test of Sphericity and the Kaiser–Meyer–Olkin (KMO) index, respectively, with a recommended KMO value of more than 0.60 [41,42,44]. Afterward, EFA was performed using the principal components analysis method with an orthogonal rotation and a maximum likelihood estimator, with an analysis of the eigenvalues to select the number of factors to be extracted. The Kaiser criterion suggested an eigenvalue greater than 1.0 for initial factors [45]. A cutoff for statistical significance of the factor loadings was 0.4 according to Comrey and Lee [46]. The reliability of SHAS was assessed using Cronbach’s alpha; a value equal to or greater than 0.70 was considered to be sufficient for reliability [47].

#### 2.1.3. Phase 3: Scale Evaluation of the Final Version of SHAS

To determine whether the final version of SHAS accurately identified significant factors for self-healing capacity in community-dwelling older adults, criterion validity was determined by evaluating the correlations between scores on the Medical Outcome Study Short Form (MOS SF-12) scale and SHAS by using CFA and Pearson’s correlation coefficient, respectively. The CFA of SHAS assessed the relevance of the questionnaire constructed through EFA. In addition, CFA was calculated using a maximum likelihood estimation method in SPSS AMOS 23 to assess the model fit according to the covariance matrix of the confirmatory data set. The same factor structure might exist in both data sets, as the model from the exploratory data set showed a good fit with the confirmatory one. The chi-square (χ^2^) value, root mean square residual (RMR), root mean square error of approximation (RMSEA), supplement of comparative fit index (CFI), and Tucker Lewis index (TLI) were calculated to assess an adequate model fixed with a value closer to or smaller than 0.06 for the RMSEA and a value closer to or higher than 0.9 for the CFI and TLI, based on the criteria by Hu and Bentler [48]. Standardized factor loadings higher than 0.5 indicated good convergent validity, and path coefficients showing significance indicated good discriminant validity.

The MOS SF-12, also filled out by participants, was included in the data collection process to allow for assessing the SHAS concurrent validity. The MOS SF-12 is two-dimensional, including physical component summary (PCS) and mental component summary (MCS) and is a self-reported scale used to measure health-related QOL [49] (Galenkamp et al., 2018). It encompasses 12 items and uses the automatic scoring system to determine the score for each dimension and overall. Higher scores indicate higher health-related QOL with a cutoff point of 50 in each PCS and MCS after norming the standardized score with an older Taiwanese population [49,50,51].

### 2.2. Sample/Participants

Using a convenience sampling method, community-dwelling older adults aged 60 and higher in Northern Taiwan were recruited from two community health centers to test the psychometric properties of the Self-Healing Assessment Scale (SHAS) between October 2018 and March 2019. Exclusion criteria were older adults who were unwilling to participate, unable to communicate in Mandarin or another Chinese dialect, had psychiatric disorders, or intellectual disabilities. The older adults who agreed to participate in the study were asked to complete the questionnaire, which comprised a 95% response rate.

### 2.3. Ethical Considerations

The institutional review board (No. 201808ES042) reviewed and approved the study protocol, questionnaire, and informed consent, which was obtained individually before the study was conducted, and the study was conducted following the Declaration of Helsinki.

## 3. Results

### 3.1. The Characteristics of the Participants

The sample of 500 older adults comprised 89 (17.8%) males and 411 (82.2%) females; the mean age was 71.76 (Std. Deviation = 6.55).

### 3.2. Content Validity and Item Analysis

The I-CVI that represented the relevance of items in evaluating the self-healing capacity was 0.89. In the item analysis, the mean and SD of SHAS ranged from 3.06 to 3.39 and 0.50 to 0.57, respectively. The scores of all items did not exceed 0.6 SD from the mean, indicating that all items were adequate. The SE Skewness and SE Kurtosis were 0.11 and 0.22, which were not greater than 1 and 7, respectively, indicating normality. In extreme groups, the t value of each item was between 13.18 and 30.09 (*p* < 0.001). The interitem correlation coefficients ranged from 0.30 to 0.65 in the homogeneity test. On the basis of the item analysis results, all items were retained at this step (Table 1 and Table 2).

### 3.3. Exploratory Factorial Analysis and Reliability

An initial 12-item questionnaire was examined through the EFA before data reduction techniques. The Bartlett’s test of Sphericity reached statistical significance (*p* < 0.05) and the KMO statistic was 0.92. SHAS showed very good internal consistency, with a Cronbach’s alpha of 0.91 in 500 complete surveys. EFA explained 65.8% of the variance, and of the three rotated factors, the first factor, accounting for 27.62% of the variance, had five items with loadings well above the cutoff score of 0.40. All of these items loaded exclusively on this factor. The internal consistency of these five items was high with a Cronbach’s alpha of 0.85, covering questions about physical and mental states. The second factor, accounting for 25.17% of the variance, had five moderate to moderately high factor loadings, ranging from 0.56 to 0.75 and a Cronbach’s alpha of 0.83, for assessing the socioeconomic and environmental status. Finally, the third factor accounted for 13.03% of the variance among the two items. Factor loading was high to moderate, ranging from 0.61 to 0.84 with an overall Cronbach’s alpha of 0.74. These items explained independent lifestyle (Table 3 and Table 4).

An orthogonal factor analysis showed that these factors were modestly correlated, as expected. The three-factor component correlation matrix is shown in Table 5.

### 3.4. Confirmatory Factorial Analysis and Concurrent Validity

Confirmatory factorial analysis was applied to the 12 items identified in the EFA data set and provided an acceptable fit based on the goodness of fit statistics: χ^2^/df = 101.11/47, *p* < 0.001, RMR = 0.01, RMSEA = 0.05, CFI = 0.98, and TLI = 0.97. Inspection revealed that none of the 12 items had quality values lower than 0.05; therefore, all of them were included. Figure 1 illustrated that the three domains could be administered individually or combined into one scale and presented to the hierarchical model with factor loading. In the model, the standardized factor loadings ranged from 0.67 to 0.79, 0.60 to 0.76, and 0.67 to 0.89 in the domain of physical and mental status, socioeconomic and environmental status, and independent lifestyle, respectively, indicating good convergent validity. Path coefficients were significant and ranged from 0.77 to 0.90, indicating good discriminant validity. Moreover, the residuals of items 1 and 2, 6 and 7, 9 and 10, and 11 and 12 were free to correlate, in which items 1, 2, and 7 belonged to the domains of physical and mental status and items 6, 9, 10, 11, and 12 belonged to the domain of socioeconomic and environmental status.

The concurrent validity was also adequate, considering that the PCS mean and SD were 48.10 and 7.43, respectively, and the MCS mean and SD were 50.33 and 7.34, respectively, in MOS SF-12 and had a significant and positive relationship with all domains. Specifically, a higher score of PCS was associated with a higher score of physical and mental status (γ = 0.45, *p* < 0.001), socioeconomic and environmental status (γ = 0.29, *p* < 0.001), independent lifestyle (γ = 0.26, *p* < 0.001), and overall scale (γ = 0.40, *p* < 0.001), and a higher score of MCS was associated with a higher score of physical and mental status (γ = 0.44, *p* < 0.001), socioeconomic and environmental status (γ = 0.45, *p* < 0.001), independent lifestyle (γ = 0.33, *p* < 0.001), and overall scale (γ = 0.45, *p* < 0.001).

## 4. Discussion

The intrinsic capacity of self-healing for pursuing wholeness in individuals has gained attention for long-term positive outcomes in gerontology [11,18,26]. The SHAS was developed to better understand the mechanism of the intrinsic self-healing capacity and its reliability and validity by using a sophisticated methodological approach.

This new instrument (SHAS) for evaluating the capacity of self-healing of community-dwelling older adults encompasses 12 items and three domains, namely, physical and mental status, socioeconomic and environmental state, and independent lifestyle. Overall, the SHAS showed a robust factor and adequate reliability and validity coefficients through both EFA and CFA, which have been widely used to assess reliability and validity [37]. The identified factors and structures for the scale were consistent with the elements reported to influence wholeness in healthy aging. Moreover, the CFA confirmed that SHAS can be employed as a tool to assess the capacity of self-healing of community-dwelling older adults in Taiwan.

The domain of independent lifestyle contains items that stress the importance of being independent, a sense of achievement to generate a sense of self-worth and wellbeing in older adults, and the need to maintain the ability of ADL in their lifespan, as expected [35,52]. This aspect of SHAS acknowledges the importance of a variety of either physical, cognitive, or social activities that aim to counter the challenges associated with the adjustment of aging to preserve independence [52,53]. Specifically, the processes between better outcomes and physical performance, cognitive ability, and social interaction are under-investigated, even considering that this cognitive decline could be slowed by encouraging physical, cognitive, and social activities [54]. For this reason, assessing older adults’ perceived understanding of what their self-healing capacities are in terms of physical, mental, social, and environmental factors (SHAS items) could address the weaknesses for developing and delivering preventive, comprehensive, and social and medical strategies to pursue individual wholeness in gerontology.

Health care providers have the responsibility to monitor the overall health and to maintain wholeness in the population. The concept of self-healing capacity was disclosed in a qualitative study in postsurgical recovery [55], and this intrinsic capacity toward healthy aging was emphasized by WHO [5], which demonstrated that self-healing capacity is the key component in aging society. This study verified that SHAS is a reliable and valid instrument for further research, and health care providers can apply it in providing preventive services, early identification of problems, interventions, and referrals to foster health and educational success. This can also include alleviating personal, family, and public concerns by identifying the problems and providing physical, psychological, emotional, and social support and interventions for community-dwelling older adults.

### Limitations

There are certain limitations to this study. First, the SHAS was developed and validated using samples in Taiwanese community-dwelling older adults; hence, there is a possible bias in terms of the national context of Taiwan. Second, the convenience sampling method used for enrollment contained an 82/18 female/male ratio and must be acknowledged as a limitation, given the report that shows a 54/46 female/male ratio of older adults in Taiwan by the Ministry of Health and Welfare [56]. Third, the independent lifestyle domain was narrowly expressed as only two items encompass it, yet just these two items were sufficient to express a scale domain with good intercorrelation; correlation with other variables during the evaluation of the factorial structure has been reported in a recent study [57]. Finally, further empirical research for confirming the factor structure of SHAS and sensitivity analysis is needed.

## 5. Conclusions

This study provided groundwork both in the definition of self-healing in healthcare and in the validity and reliability of SHAS. Considering the feasibility of SHAS, a short questionnaire with a 12-item self-reflection capacity of self-healing in community-dwelling older adults can be assessed through cross-sectional, longitudinal, or interventional studies to examine the causal relationships or outcomes of the factors of SHAS and a sense of wholeness in older adults’ later lifespan. From a practical perspective, this newly developed measurement of self-healing capacity is suitable for further expansion, specifically, in evaluating community health promotion intervention programs, reducing the gap between a lifespan and healthy lifespan, and improving older adults’ QOL.

## Figures and Tables

**Figure 1 healthcare-09-00484-f001:**
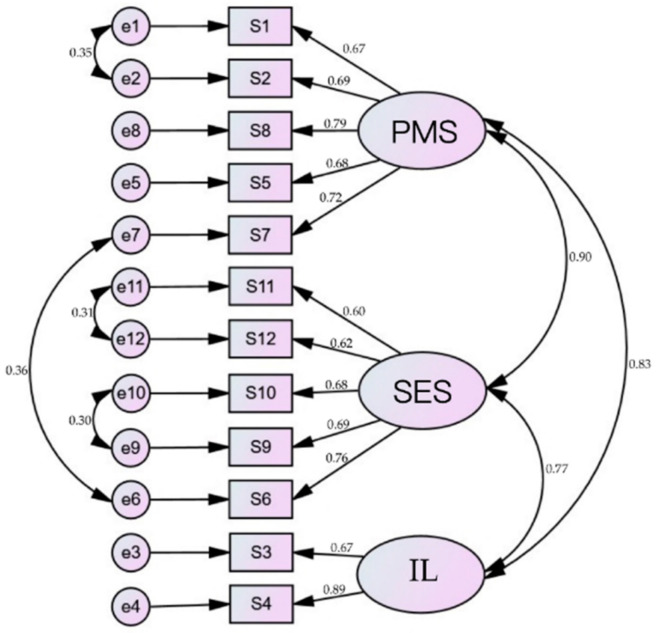
Community-dwelling older adults’ self-healing model. Confirmatory factor analysis based on 12 items and three factors. Note. PMS = physical and mental state; SES = socioeconomic and environmental state; IL = independent lifestyle.

**Table 1 healthcare-09-00484-t001:** Item analysis.

Questionnaire Items	Mean	Std. Deviation	Skewness		Kurtosis		t	95% Confidence Interval	
	Statistic	Statistic	Statistic	Std. Error	Statistic	Std. Error		Lower	Upper
Item 1.	3.16	0.51	0.25	0.11	0.38	0.22			
Item 2.	3.06	0.54	0.05	0.11	0.45	0.22	−13.18 *	−0.64	−0.47
Item 3.	3.39	0.55	−0.17	0.11	−0.45	0.22	−30.09 *	−0.94	−0.82
Item 4.	3.23	0.51	0.28	0.11	−0.17	0.22	−18.50 *	−0.75	−0.60
Item 5.	3.15	0.51	0.04	0.11	1.28	0.22	−14.59 *	−0.65	−0.50
Item 6.	3.14	0.51	0.05	0.11	1.48	0.22	−17.58 *	−0.84	−0.67
Item 7.	3.15	0.51	0.14	0.11	0.94	0.22	−19.40 *	−0.87	−0.71
Item 8.	3.10	0.57	0.01	0.11	−0.02	0.22	−21.75 *	−0.98	−0.82
Item 9.	3.11	0.56	−0.10	0.11	0.69	0.22	−15.38 *	−0.77	−0.60
Item 10.	3.15	0.50	0.19	0.11	1.07	0.22	−15.37 *	−0.70	−0.55
Item 11.	3.24	0.53	−0.02	0.11	0.60	0.22	−18.99 *	−0.82	−0.67
Item 12.	3.22	0.51	0.21	0.11	0.50	0.22	−17.53 *	−0.76	−0.60

Note: *n* = 500; * *p* < 0.05.

**Table 2 healthcare-09-00484-t002:** Item correction matrix.

Item	1	2	3	4	5	6	7	8	9	10	11	12
1	-											
2	0.65 **	-										
3	0.38 **	0.38 **	-									
4	0.53 **	0.53 **	0.60 **	-								
5	0.49 **	0.51 **	0.43 **	0.53 **	-							
6	0.46 **	0.45 **	0.39 **	0.52 **	0.50 **	-						
7	0.44 **	0.47 **	0.36 **	0.50 **	0.42 **	0.67 **	-					
8	0.54 **	0.55 **	0.43 **	0.55 **	0.50 **	0.55 **	0.65 **	-				
9	0.43 **	0.39 **	0.30 **	0.46 **	0.47 **	0.52 **	0.43 **	0.47 **	-			
10	0.37 **	0.36 **	0.34 **	0.46 **	0.40 **	0.50 **	0.48 **	0.47 **	0.63 **	-		
11	0.34 **	0.36 **	0.32 **	0.39 **	0.34 **	0.45 **	0.44 **	0.40 **	0.43 **	0.48 **	-	
12	0.33 **	0.36 **	0.33 **	0.45 **	0.36 **	0.47 **	0.45 **	0.41 **	0.43 **	0.47 **	0.57 **	-

Note: *n* = 500; ** *p* < 0.01.

**Table 3 healthcare-09-00484-t003:** Rotated factors for principal components analysis of SHAS.

Questionnaire Items	Factor Loading		
	1	2	3
*Factor 1: physical and mental state*			
1. I perceive my self-healing as good	0.78		
2. I have a good health status	0.78		
8. I feel a positive vitality everyday	0.69		
5. I have a good management of my health status	0.60		
7. I am in a pleasant mood most of the times	0.59		
*Factor 2: socioeconomic and environmental status*			
11. My economic status is good		0.75	
12. My community living environment is good		0.74	
10. I have good social support		0.73	
9. My interpersonal relationship is good		0.65	
6. I can reach a peaceful mindset		0.56	
*Factor 3: independent lifestyle*			
3. I have good daily living functions			0.84
4. I have good lifestyle			0.61
Eigenvalue	3.32	3.02	1.56
% of variance	27.62	25.17	13.03
Cumulative %	27.62	52.80	65.83

Note: *n* = 500.

**Table 4 healthcare-09-00484-t004:** Reliability analysis using Cronbach’s α, M, and SD.

Scale	Components	Cronbach’s α	Mean	Std. Deviation
SHAS		0.91	38.10	4.45
	Physical and Mental State	0.85	15.62	2.07
	Socioeconomic and Environmental State	0.83	15.86	2.01
	Independent Lifestyle	0.74	6.62	0.94

Note: *n* = 500.

**Table 5 healthcare-09-00484-t005:** Component correlation matrix of 3 factors.

Components	Physical and Mental State	Socioeconomic and Environmental State	Independent Lifestyle
Physical and Mental State	-		
Socioeconomic and Environmental State	0.70 **	-	
Independent Lifestyle	0.66 **	0.57 **	-

Note: *n* = 500; ** *p* < 0.01.

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
