# Peer review of "Psychometric Properties of the Self-Healing Assessment Scale for Community-Dwelling Older Adults"

_healthcare, 2021, doi:10.3390/healthcare9040484_

Round 1

Reviewer 1 Report

Thank you for the opportunity to review this paper. This paper is interesting and important topic to develop and examine the self-healing assessment scale (SHAS). I enjoyed reading the paper and I hope you find the following comments helpful.

Introduction is clear.

2.1.1. Phase 1: item development of the initial version of SHAS

Did you to literature review to define the concept self-healing? You are writing: The definitions of self-healing were synthesized from our knowledge of the literature regarding the ontology of self-healing and panels. Is it possible to describe used search terms and databases? This will improve the phase 1 reliability.

How panels were conducted in practice? By email, by interviews? Please clarify.

2.2. Sample/Participants

Recruitment process needs more detailed description

Discussion is quite narrow. What is the value of this instrument of SHAS in the future to evaluate and support community-dwelling older adults’ self-healing capacity?

Author Response

Dear Reviewer:

      We revised the content according to your comments.

Sincerely, Heng-Hsin

Reviewer 2 Report

Dear Authors,

Thank you for giving me the possibility to review your manuscript “Psychometric Properties of the Self-Healing Assessment Scale for Community-dwelling Older Adults”. I read your manuscript with interest. Overall, I am satisfied with it. Below you will find my suggestions for improvement.

Abstract

The abstract gives a good representation of your study.

Introduction

I have no comments on the introduction.

Methodology

Section 2.1.1 of Methodology start with a paragraph describing a lot of definitions of (self-)healing. This paragraph does not belong in Methodology. I suggest adding some of this to the introduction. The second paragraph starts with ‘Self-healing was expected to be a difficult concept to define and discuss,……. Explain this in more detail.

Page 4, two times a Cronbach’s alpha of .07 was mentioned. I think this should be .70.

Page 4, 2.1.3. Third line “a criterion validity of the questionnaire was conducted’. This is not good English. Please rephrase.

Page 5. 2.2. Taiwan. Where exactly in Taiwan? How did you recruit the community-dwelling older adults? Can you add a response rate?

Page 5. 2.2. Describe the characteristics of the participants at Results.

Results

Table 1. Write the abbreviations in full below the table.

Page 6. Change “These items explain” in “These items explained”

Discussion

Page 8. Change “This new instrument of SHAS” in “This new instrument (SHAS)”.

Limitations. I agree with the limitations of your study. Please be more specific about “further empirical research”. What do you suggest?

Elaborate more on the practical usefulness of the SHAS.

Author Response

(The authors gave the same response as above.)

Round 2

Reviewer 2 Report

The authors addressed my comments satisfactorily. I suggest a few more minor adjustments.

Page 4 (2.1.3.) Change the sentence “…., criterion validity was undertaken to test the questionnaire in evaluating the internal consistency and correlations between scores…………”

Criterion validity doesn’t refer to internal consistency of a scale, so I suggest to change the sentence in ““…., criterion validity was determined by evaluating the correlations between scores…………”

Page 9 Change the sentence “This study verified that SHAS is a reliable instrument for………” in “This study verified that SHAS is a reliable and valid instrument for………”

Author Response

     We revised the manuscript according to the reviewer's comments as an attachment.

Sincerely, Heng-Hsin 
